# Adverse events associated with the delivery of telerehabilitation across rehabilitation populations: A scoping review

Thomas Yau[1], Josh Chan[2], McKyla McIntyre [3,4], Damanveer Bhogal[1], Angie Andreoli[3], Carl Froilan D. Leochico [1,5,6], Mark Bayley[1,3,4,7,8,9], Ailene Kua [3,7], Meiqi Guo [3,4,7‡], Sarah Munce [10,11‡*]

1 Temerty Faculty of Medicine, University of Toronto, Toronto, Ontario, Canada, 2 Western University, London, Ontario, Canada, 3 Toronto Rehabilitation Institute-University Health Network, Toronto, Ontario, Canada, 4 Division of Physical Medicine and Rehabilitation, Department of Medicine, Faculty of Medicine, University of Toronto, Toronto, Ontario, Canada, 5 Department of Physical Medicine and Rehabilitation, St. Luke's Medical Center, Global City and Quezon City, Philippines, 6 Department of Rehabilitation Medicine, Philippine General Hospital, University of the Philippines Manila, Manila, Philippines, 7 KITE Research Institute, Toronto Rehabilitation Institute-University Health Network, Toronto, Ontario, Canada, 8 Department of Occupational Science and Occupational Therapy, University of Toronto Rehabilitation Sciences Institute, University of Toronto, Toronto, Ontario, Canada, 9 Institute of Health Policy, Management and Evaluation, University of Toronto, Toronto, Ontario, Canada, 10 Rehabilitation Sciences Institute, University of Toronto, Toronto, Ontario, Canada, 11 Bloorview Research Institute, Holland Bloorview Kids Rehabilitation Hospital, Toronto, Ontario, Canada

☺ These authors contributed equally to this work.
‡ MG and SM are co-senior authors on this work.
* sarah.munce@uhn.ca

**Data Availability Statement:** All relevant data are within the manuscript and its Supporting information files.

## Abstract

### Objective

This scoping review aimed to map existing research on adverse events encountered during telerehabilitation delivery, across rehabilitation populations. This includes identifying characteristics of adverse events (frequency/physical/non-physical, relatedness, severity) and examining adverse events by different modes of telerehabilitation delivery and disease states.

## Introduction

Telerehabilitation, a subset of telemedicine, has gained traction during the COVID-19 pandemic for remote service delivery. However, no prior scoping review, systematic review, or meta-analysis has identified and summarized the current primary research on adverse events in telerehabilitation. Understanding adverse events, such as falls during physiotherapy or aspiration pneumonia during speech therapy, is crucial for identifying limitations and optimizing delivery through risk mitigation and quality indicators. This understanding could also help to improve the uptake of telerehabilitation among clinicians and patients. This review addresses this gap by summarizing published literature on adverse events during telerehabilitation.

**Funding:** Author TY received funding through a Toronto Rehabilitation Institute summer research position, funded by a donation to the University Health Network.

**Competing interests:** The authors have declared that no competing interests exist.

## Methods

The review followed the Joanna Briggs Institute framework and adhered to the Preferred Reporting Items for Systematic Reviews and Meta-Analyses Extension for Scoping Reviews guidelines. The review protocol was registered and published on Open Science Framework. A comprehensive search across multiple databases (MEDLINE ALL/EMBASE/APA PsycINFO/CENTRAL/CINAHL) was conducted. Screening, extraction, and synthesis were performed in duplicate and independently. Data extraction followed the Template for Intervention Description and Replication framework and also involved extraction on authors, publication year (pre- or post-COVID), population, sample size, and modes of telerehabilitation delivery (asynchronous, synchronous, hybrid). For synthesis, data were summarized quantitatively using numerical counts and qualitatively via content analysis. The data were grouped by intervention type and by type of adverse event.

## Inclusion criteria

This scoping review included qualitative and quantitative studies published between 2013–2023, written in English, and conducted in any geographic area. All modes of telerehabilitation delivery were included. Systematic reviews, meta-analyses, commentaries, protocols, opinion pieces, conference abstracts, and case series with fewer than five participants were excluded.

## Results

The search identified 11,863 references, and 81 studies were included in this review with a total of 3,057 participants (mean age:59.3 years; females:44.6%). Modes of telerehabilitation delivery (whether asynchronous, synchronous or hybrid) used in the studies included videoconferencing (52), phone calls (25), text messaging (4), email (6), mobile apps (10), and internet-based virtual reality systems (3). A total of 295 adverse events occurred during 84,534 sessions (0.3%), with the majority being physical (e.g., falls or musculoskeletal pain), non-serious/non-severe/mild, and unrelated to (i.e., not caused by) to the telerehabilitation provided.

## Conclusions

From the 81 included studies, telerehabilitation was delivered with related adverse events being rare, and mostly characterized as mild/non-severe. A comparable occurrence of adverse events (~30%) was found between asynchronous and synchronous telerehabilitation studies. When categorized by disease type, cardiac telerehabilitation studies had the most frequent adverse events. Detailed reporting of telerehabilitation interventions and adverse event characteristics is recommended for future studies (i.e., use of TIDieR reporting guidelines). Telerehabilitation has the potential to make rehabilitation services more accessible to patients; however, more evidence on the safety of telerehabilitation is needed.

## Introduction

Telerehabilitation is a subset of telemedicine connecting rehabilitation providers and patients at a distance [1]. Telerehabilitation can also provide services to those who would not normally be able to access traditional rehabilitation, such as those living in remote communities or patients with disabilities which hinder participation in in-person sessions [2], assuming that appropriate internet connections are available. The convenience of telerehabilitation may lead to decreased travel expenses for participants and higher attendance rates for individuals with other life commitments [3]. Multiple systematic reviews have shown the effectiveness of telerehabilitation; for instance, a study by Dias et al. found high-quality evidence that telerehabilitation was not different from other interventions for adults with physical disabilities in terms of long-term improvements in pain, physical function, and quality of life [4–8]. However, there remain questions about potential limitations of telerehabilitation, in particular its safety compared to in-person rehabilitation [9]. Due to the remote nature of telerehabilitation, patients cannot receive immediate physical assistance from rehabilitation providers if they experience an adverse event, which are defined as "negative consequences of care that result in unintended injury or illness which may or may not have been preventable" [10, 11]. For instance, they may include falls during physiotherapy or aspiration pneumonia due to speech language pathology swallowing assessments [12, 13]. There is a paucity of research surrounding the patient safety of telerehabilitation, potentially contributing to its limited uptake among clinicians and patients [14]. The rationale for the current review is as follows: while many individual studies include safety data, there exists a research gap as there has yet to be any synthesis of the existing literature that summarizes the currently available research on adverse events related to telerehabilitation. There has been a prior scoping review on measures to ensure safety during telerehabilitation for patients with stroke, specifically, but the current review differs as it focuses on adverse events and encompasses all health/chronic conditions that could be served by telerehabilitation [15]. This scoping review aimed to conduct a systematic search of published literature on adverse events during the delivery of telerehabilitation, across rehabilitation populations, and map out the extent of existing research. This included identifying characteristics of adverse events (frequency, physical versus non-physical, relatedness to telerehabilitation, severity) and examining adverse events for different modes of telerehabilitation delivery and disease states. The World Health Organization (WHO) recognizes patient safety as a global health priority, and notes that investing in patient safety is important for health outcomes, cost reduction related to patient harm, and health system efficiency [16]. It is important to understand adverse events associated with telerehabilitation delivery, so that safety precautions and risk-mitigation measures can be thoughtfully planned and implemented, to optimize telerehabilitation's uptake and delivery. Knowledge of the safety of telerehabilitation can help patients make more informed decisions, aid in clinical and funder decision-making and inform safety quality indicators for telerehabilitation.

## Methods

This review adhered to the Joanna Briggs Institute (JBI) methodological framework for scoping reviews, which provides guidance on the outline of the review, inclusion criteria (i.e. Population (or participants)/Concept/Context), search strategy, extraction, presenting and summarizing the results, and any potential implications of the findings for research and practice [17]. The reporting of the scoping review adhered to the Preferred Reporting Items for Systematic Reviews and Meta-Analyses Extension for Scoping Reviews (PRISMA-ScR) guidelines, to ensure all the components of a high-quality scoping review were completed, and a

filled checklist is viewable in the S1 Appendix [18]. Our team included members with extensive experience in scoping reviews and telerehabilitation.

## Protocol and registration

The protocol was registered and published on Open Science Framework on June 26, 2023 (Registration DOI: https://doi.org/10.17605/OSF.IO/C3ZHQ).

## Eligibility criteria

Various study designs were considered in this scoping review (e.g., experimental, quasi-experimental (quasi), observational, qualitative, mixed, and multiple methods). However, systematic reviews, meta-analyses, commentaries, protocols, opinion pieces (editorials), abstracts from conferences, and case series of <5 participants were not included. Studies were limited to those published between 2013–2023, because a study by Zheng et al. found that 2013 was the start of a more significant development period of telerehabilitation, with only a few papers on telerehabilitation published prior [19]. Additionally, the year 2013 marked the emergence of video communication technologies such as Zoom or Google Hangout that are commonly used in telerehabilitation today, which ensures that the review's results are relevant to the current practice of telerehabilitation [20]. Studies had to be written in the English language but could be from any geographic area. All modes of delivery for telerehabilitation (asynchronous, synchronous, or hybrid) were eligible.

## Search strategy

Search strategies were developed by a librarian with experience searching the health sciences literature and conducting systematic and scoping reviews. The following databases were searched on the Ovid platform: MEDLINE ALL, EMBASE, APA PsycINFO, and Cochrane Central Register of Controlled Trials (CENTRAL). The Cumulative Index to Nursing and Allied Health Literature (CINAHL) database was searched on the EBSCOhost platform. An initial strategy was created in MEDLINE ALL and sent to the team for review. Once the test strategy for MEDLINE ALL was agreed upon, the librarian sought out a volunteer librarian to provide a Peer Review of Electronic Search Strategies (PRESS) review.

The MEDLINE ALL search strategies were translated using the command language, controlled vocabulary, and appropriate search fields for each database and search platform. Search terms included Medical Subject Headings (MeSH), EMTREE terms, American Psychological Association thesaurus terms, and CINAHL headings and text words to capture concepts and synonyms of telerehabilitation and adverse events. Results were limited to the English language and the publication period from 2013 to present. The full MEDLINE ALL search strategy can be viewed in S2 Appendix.

## Study/Source of evidence selection

All identified citations were imported into EndNote, a reference management tool, to remove duplicates. They were then transferred into Covidence (https://www.covidence.org/), a web-based reference manager software. All rounds of screening were completed in duplicate and independently. After completing a pilot test, titles and abstracts were screened. Sources that met the inclusion criteria were retrieved in full. A list of excluded studies can be seen in S3 Appendix. This was then followed by a round of screening based on full texts.

## Data extraction

Reviewers TY, JC, DB, AA, CL, and MG independently piloted the extraction form with a random sample of studies and made necessary revisions. Each study was abstracted in duplicate by two different reviewers. TY completed the extraction of all studies, and JC, DB, AA, CL, and MG completed the extraction in duplicate. Extractor details can be viewed in S4 Appendix. The results from the two reviewers were compared, and conflicts were resolved by discussions between TY and MG. The data extracted followed the Template for Intervention Description and Replication (TIDieR) framework, including name of intervention, objective/rationale, materials used, procedures, provider, specific mode/s of telerehabilitation delivery (full telerehab vs in-person hybrid, synchronous: videocall, phone call, instant messaging, web-based such as using either virtual reality or augmented reality; asynchronous: text/ audio/ video messaging, e-mails, on-demand resources; or hybrid: combination of any synchronous and asynchronous methods), location of therapist and patient, period of time, number of sessions, schedule, duration, intensity/dose, tailoring, modification over the course of study, and adherence [21]. It also included specific details including authors, year of publication (before or after the COVID-19 pandemic with pre-COVID defined as data collection starting before March 11, 2020 according to the World Health Organization) [22], population and sample size (including number of patients in each arm and total sample size), age and sex of participants, population type, study design, materials used, and outcome measures (adverse events including type, expected versus unexpected, related versus unrelated, and description). The extraction form can be viewed in S5 Appendix. Quality/risk of bias assessment was not completed as this was not the purpose of a scoping review. All possible data that could be extracted from the included studies were extracted; there were no missing data.

## Data analysis and presentation

Data from this scoping review were summarized quantitatively using numerical counts and qualitatively via content analysis, based on best practices for reporting of scoping reviews [23]. The data were grouped by intervention type and by adverse event type (physical, social, psychological) and analyzed/coded manually. Numerical counts and content analysis were used to reveal trends in the data such as the most common method of telerehabilitation, the health condition with the most adverse events, and the frequency of different types of adverse events. Synthesis occurred in duplicate and independently.

## Results

### Study/Source of evidence selection

A total of 11,863 references were identified from the initial search. After removing 3,506 duplicates via EndNote, a total of 8,357 references remained. Next, 833 other duplicates were identified and removed after importing all records into Covidence, leaving a total of 7,524 studies for title and abstract screening. Reviewers TY, JC, and DB independently piloted a random subsample of titles and abstracts to identify any revisions of the inclusion/exclusion criteria. After revision, title and abstract screening commenced and 7,346 studies were found to be irrelevant, leaving 178 studies for full-text screening. Full-text screening identified 97 studies for exclusion, leaving 81 studies remaining for extraction. The results of the search and study inclusion process are illustrated in a PRISMA-ScR flow diagram (see S6 Appendix).

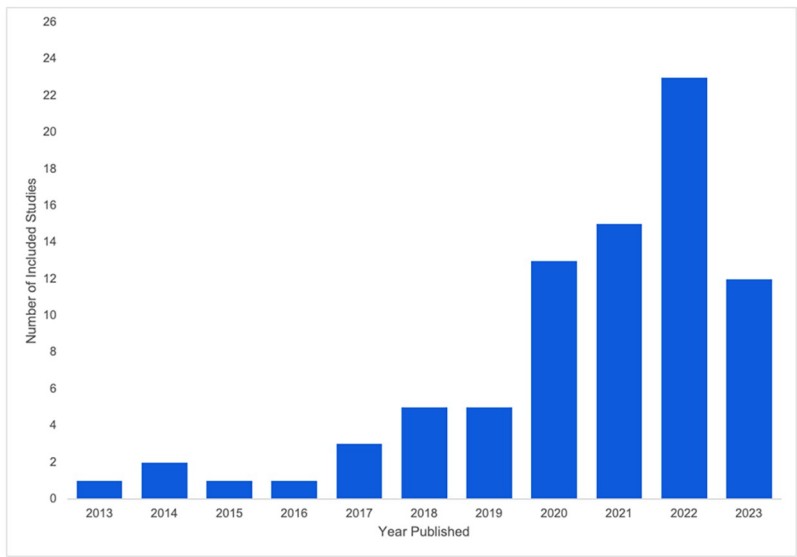

**Fig 1. Included studies by publication year.**

## Study characteristics

The total number of included studies was 81, of which 43 (53.1%) were published post-COVID. The distribution of studies by publication year is shown in Fig 1, demonstrating the increase in articles on telerehabilitation-related adverse events from 2020 onwards.

## Population and sample size

From the 81 studies, there were a total of 3,057 participants in the intervention groups, with the mean, median, minimum, and maximum number of participants being 37.7, 17, 4, and 425, respectively. There were a total of 38 studies that had control groups, with a total of 1,486 participants. The mean, median, minimum, and maximum number of controls were 39.1, 24, 5, and 425, respectively. Thus, for the studies that did have controls, the number of participants in the intervention and control groups seemed relatively even.

The mean age was 59.3 years, not including a study involving primary and secondary school children as mean age was not reported. Almost half (44.5%) of the participants in the included studies were female. The number of articles from each country of origin can be seen in Fig 2.

The included studies comprised of participants with various disease conditions. A full breakdown can be seen below and visualized in Fig 3.

Cardiac (18 studies, 22.2%)—cardiac/cardiovascular disease, 6 [24–29]; heart failure, 5 [30–34]; coronary artery/heart disease, 3 [35–37]; transcatheter aortic valve implantation (TAVI), 2 [38, 39]; advanced combined cardiopulmonary disease, 1 [40]; pediatric heart disease, 1 [41].

Respiratory (17 studies, 20.9%)—COVID-19, 8 [42–49]; COPD, 3 [50–52]; lung transplant, 3 [53–55]; chronic respiratory disease, 1 [56]; lung cancer, 1 [57]; severe cystic fibrosis, 1 [58].

Stroke (13 studies, 16.0%)—stroke, 10 [59–68]; upper extremity dysfunction/paresis from stroke, 2 [69, 70]; post-stroke aphasia, 1 [71].

Cancer (8 studies, 9.9%)—esophageal/esophagogastric cancer, 3 [72–74], unspecified cancer, 2 [75, 76]; glioma, 1 [77]; hematological cancer, 1 [78]; pediatric cancer, 1 [79].

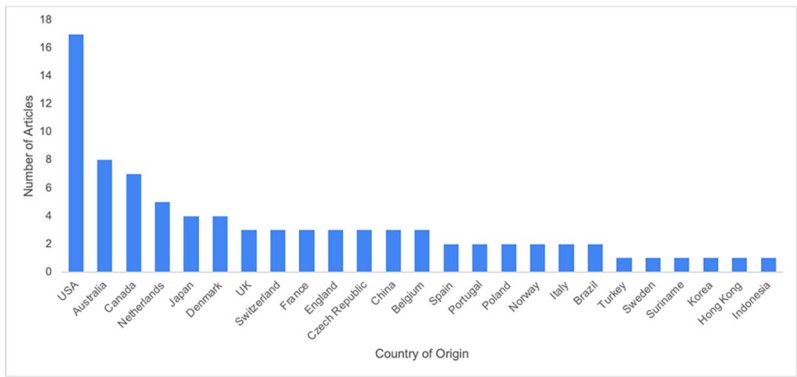

**Fig 2. Included studies by country of origin.**

Neurological (16 studies, 19.8%)—Parkinson's disease, 8 [80–87]; cerebral palsy, 2 [88, 89]; multiple sclerosis, 2 [90, 91]; dementia, 1 [92]; mild cognitive impairment, 1 [93]; mild traumatic brain injury, 1 [94]; unspecified neurologic diseases, 1 [95].

Musculoskeletal (5 studies, 6.2%)—anterior cervical discectomy and fusion, 1 [96]; burn injury, 1 [97], chronic musculoskeletal pain, 1 [98]; inflammatory myopathies, 1 [99], total knee replacement, 1 [100].

Others (4 studies, 4.9%)—geriatric, 1 [101]; overweight and obese, 1 [102]; sarcoidosis, 1 [103]; HIV, 1 [104].

## Study design

From the 81 included studies, a total of 84,534 telerehabilitation sessions were conducted. For study design, 36 studies were quasi-experimental [29, 31, 33, 35, 37, 39–44, 48, 50–52, 57, 58, 61, 63–67, 69, 72, 78–80, 82, 83, 85, 87, 92, 99–101], 12 were mixed methods [38, 54, 55, 62, 73, 76, 81, 84, 89, 93, 95, 96], 2 were qualitative [59, 68], and 3 were observational [47, 49, 75]. There were 28 randomized controlled trials included [24–28, 30, 32, 34, 36, 45, 46, 53, 56, 60, 70, 71, 74, 77, 86, 88, 90, 91, 94, 97, 98, 102–104].

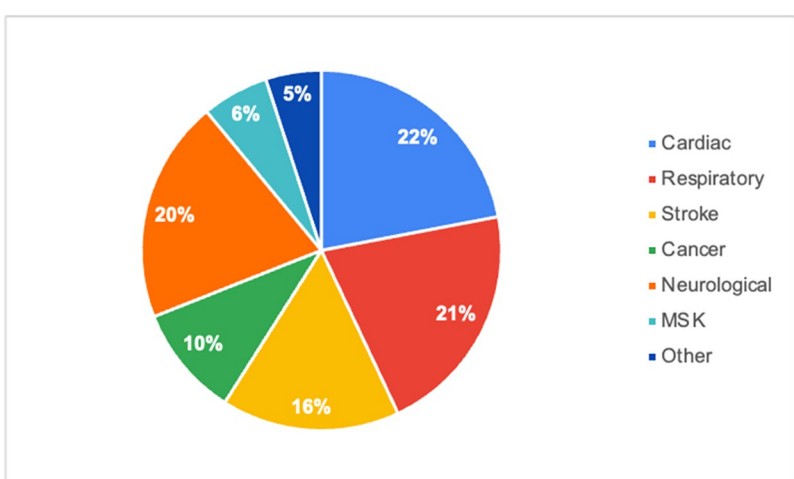

**Fig 3. Included studies by disease type.** MSK = musculoskeletal.

## Intervention characteristics

There were 63 studies that were fully telerehabilitation-based (i.e., patients and therapists in separate physical locations) [25–27, 29–32, 34–37, 40–43, 45–50, 52–55, 57–59, 61, 63, 64, 66, 68–71, 73, 76–85, 87–98, 100–102, 104], and 18 were hybrid (i.e., with an in-person component) [24, 28, 33, 38, 39, 44, 51, 56, 60, 62, 65, 67, 72, 74, 75, 86, 99, 103]. There were 46 studies that used synchronous (real-time) methods of delivering telerehabilitation [24, 29–32, 34, 35, 38–44, 46–52, 55–57, 62–65, 69, 71, 74–76, 81–85, 87, 89, 92, 94, 95, 97, 101, 102], 30 studies that used asynchronous (store-and-forward) methods [25–28, 33, 36, 37, 53, 54, 58–61, 66, 70, 72, 73, 77, 78, 86, 88, 90, 91, 93, 96, 98–100, 103, 104], and 5 studies that were hybrid [45, 67, 68, 79, 80]. On average, sessions were provided 2.68 times per week, for 10.26 weeks, and for 46.52 minutes per session. Various communication methods were used: videoconferencing (52 studies) [24, 27, 29–32, 34, 35, 38–41, 43–52, 55–57, 62–65, 67–69, 71, 73, 75, 76, 79–85, 87, 89, 92, 94, 95, 97, 101–103], phone calls (25) [25, 26, 28, 33, 36, 37, 53, 59, 60, 70, 72, 73, 75–79, 81, 86, 88, 93, 96, 98, 100, 104], mobile application (10) [34, 42, 53, 54, 59, 66, 74, 76, 99, 103], email (6) [54, 58, 72, 73, 77, 91], Internet-based virtual reality (VR) systems (3) [61, 67, 93], and text messaging (4) [36, 58, 79, 98].

Studies employed different healthcare providers, with the majority being physiotherapists (61 studies), followed by nurses (12), physicians (11), exercise physiologists/sport exercise and rehabilitation researchers/certified sports medicine exercise trainers (6), occupational therapists (4), speech-language pathologists (2), kinesiologists (1) and an adapted physical activity coach (1). One study had an unspecified multidisciplinary team with therapy staff and allied health professionals, while another had an unspecified type of licensed therapist. In some studies, other allied health professionals also provided health counseling/interventions. These included clinical pharmacists in 1 study, respiratory therapists in 1 study, nutritionists in 1 study, and psychologists in 2 studies.

The interventions included different types of therapy and exercises: strength/resistance in 45 studies, aerobic/endurance/cardiovascular in 43 studies, balance in 18 studies, stretching in 9 studies, walking in 7 studies, respiratory function training/breathing exercises in 8 studies, cognitive rehabilitation in 4 studies, speech language therapy/swallowing training in 3 studies, yoga in 2 studies, neurological rehabilitation for upper limb motor recovery in 1 study, and dance in 1 study. There were also a few studies that did not define the telerehabilitation intervention as any of these categories. These included studies involving Virtual Reality (VR) exergames, transcranial direct current stimulation with tracking training therapy, and pediatric rehabilitation (enhancing motor development and environmental enrichment).

While most studies used common hardware like a phone/tablet/laptop/video camera, a few used more technologically advanced tools for telerehabilitation. There were 6 studies that used virtual reality systems. Studies involving VR exergames monitored patients' movements via gesture tracking using a force plate, inertial motion trackers, infrared sensors, or a Kinect sensor. Certain devices used consoles and a range of controllers for gross and fine movement, as well as tangible user interfaces for goal-based activities. One study employed the use of transcranial direct current stimulation equipment. Another set up external kiosks with computers and webcams in the nearby community due to the low availability of affordable home broadband connections. Commercially available game systems, such as the Xbox Dance Mat or NintendoVR Wii Fit, were part of a few interventions.

## Adverse events

A total of 295 adverse events occurred from the 84,534 telerehabilitation sessions, representing 0.3%. Of the 81 included studies, 59 reported no adverse events.

**Table 1. Adverse events by severity.**

| Severity | Number of Adverse Events |
|---|---|
| **Non-injury** | 1 |
| **Non-serious, non-severe, mild** | 129 [b] |
| **Minor** | 50 [a],[b] |
| **Moderate** | 6 |
| **Serious or severe** | 20 [b] |
| **Severity not defined** | 62 [a] |

[a] = one study with adverse events in this category did not report the total number of adverse events.

[b] = one study with adverse events in this category only reported the number of participants per severity.

There were 22 studies (10 quasi, 5 mixed, and 7 RCT) that identified adverse events, involving 1,241 participants over 41,186 sessions.

Of the 81 included studies, 40.0% of total adverse events (n = 118) were obtained from those that employed synchronous telerehabilitation. Among those studies using synchronous telerehabilitation, 28.2% indicated some adverse events. In contrast, 60.0% of total adverse events (n = 177) were obtained from studies that employed asynchronous telerehabilitation. Of those studies using asynchronous telerehabilitation, 30.0% indicated some adverse events. No adverse events were reported from those that employed hybrid telerehabilitation.

The severity of the adverse events was mostly classified as non-serious, mild, or minor by the study authors (Table 1).

There was a broad range of descriptors used to identify the degree to which the adverse event was related to the telerehabilitation intervention, as opposed to other factors; many of these were not defined by any specific criteria. Only 15 of the 295 total adverse events were characterized by study authors as related, 2 probably related, 8 possibly related, 1 potentially related, 5 unlikely, 147 unrelated, and 94 undefined. There were 2 studies that only reported the number of participants per level of relatedness, rather than the specific number of adverse events. Another study reported a description of the adverse events but without a numerical count or report of relatedness.

Almost all the adverse events were physical. The majority were pain-related (musculoskeletal pain/strain or headache), but also included fatigue, falls, dizziness, and cardiac-related adverse events (e.g., chest discomfort, palpitations, angina, tachypnea, etc.). Only 3 adverse events were non-physical. For instance, one participant felt concerned about the possibility of falling, another participant had an unrelated adverse event of anxiety, and a patient with depression felt demotivated when asked to complete a measure of mood [48, 64, 93].

When categorized by disease type, cardiac rehabilitation studies had the most frequent adverse events, with 92 adverse events across 6 studies, including fatigue, palpitations, angina, diaphoresis, dyspnea, and syncope [24, 26, 28, 29, 31, 32]. Studies on people with Parkinson's disease had the second most, with 75 adverse events across 3 studies, such as pain, neuropathy, loss of balance, dizziness, falls, and increased awareness of hand tremors [82, 83, 86]. Studies on multiple sclerosis were third, with 60 adverse events from 1 study, including falls and skin reactions [91].

## Discussion

There has yet to be any synthesis of the existing literature that summarizes the currently available research on adverse events related to telerehabilitation. Thus, this scoping review aimed

to conduct a systematic search of published literature on adverse events during the delivery of telerehabilitation, across rehabilitation populations, and map out the extent of existing research. This included identifying characteristics of adverse events (frequency, physical versus non-physical, relatedness to telerehabilitation, severity) and examining adverse events for different modes of telerehabilitation delivery and disease states.

From the 81 included studies, 295 adverse events occurred during 84,534 sessions, with the majority being physical (e.g., musculoskeletal pain, falls, dizziness), non-serious/non-severe/mild, and unrelated/not caused by the telerehabilitation provided.

Using March 11, 2020, as a marker of pre- versus post-COVID, 53.0% of studies were classified as post-COVID. This may appear to oppose the expected higher number of post-COVID studies following the increase in telerehabilitation usage post-COVID. However, our range of years (2013–2023) included more pre-COVID years, and the accumulated number of pre-COVID studies was substantial. Examining the articles published by each year, there was an increase in articles which reported on adverse events during telerehabilitation delivery from the year 2020 to the present day, indicating the rise in popularity of telerehabilitation following the COVID-19 pandemic. This is in alignment with other studies reporting a general increase in telehealth usage during/following the COVID-19 pandemic [105, 106].

The mean age was 59.3, and almost half (44.5%) of the participants were female. This was a notable finding as there was no age limit in the exclusion criteria of this scoping review. There were only four included studies that involved pediatric populations [41, 79, 88, 89]. It is possible that the limited number of pediatric studies is due to younger children being less likely to participate in telerehabilitation due to the need for a caregiver to be available to accompany them for sessions, or due to their limited ability to remain stationary for the video-to-video format of telerehabilitation. However, it is noteworthy that none of the four pediatric studies identified any adverse events during the delivery of telerehabilitation. Furthermore, of the 81 included studies, only 12 studies (14.8%) reported race. Analyzing adverse events through an equity lens can help to prevent bias and inequities, and lead to corrections of root causes [107].

From the 81 included studies, there were 28 randomized controlled trials. There were a small number of studies characterized as mixed methods (12 studies) and qualitative (2 studies). A greater number of qualitative studies would increase the understanding of patient and clinician lived experiences and perspectives on telerehabilitation and adverse events.

There were a range of different interventions, using different telerehabilitation modalities and providing different types of rehabilitation (mostly aerobic and resistance training). While most studies were entirely telerehabilitation-based, 18 studies included an in-person component. Oftentimes, a few initial sessions were completed in-person so that the therapists could complete an initial physical assessment, or participants could familiarize themselves with the rehabilitation intervention or tools. Videoconferencing and phone calls were the most commonly used methods of telerehabilitation. However, a few studies included more technologically advanced tools, such as virtual reality exergames with gesture tracking or telerehabilitation devices with consoles and controllers. The COVID-19 pandemic accelerated innovations in telerehabilitation, with clinicians and researchers aiming to better understand ways to deliver telerehabilitation in a safe and effective way [108]. The development of new technologies may pave the way to more accessible and safer telerehabilitation, but it is important to test and monitor for safety when implementing new technologies [109]. Lastly, a range of therapists provided telerehabilitation, but physiotherapists were the most common. Multiple physiotherapy organizations across Canada have released guidelines for telerehabilitation delivery, indicating an increased interest in telerehabilitation within the field, which is in alignment with our findings [110, 111].

For the large total number of sessions across the included studies (81 studies), there was a low number of adverse events (295 adverse events/84,534 sessions, 0.35%). While the studies with adverse events represented only 27.0% of the total number of included studies, the number of sessions covered by those studies was substantial. Specifically, nearly half of the total number of sessions from all the included studies were from those studies with adverse events (41,186 sessions/84,534 total sessions). This may indicate that studies that held more sessions (frequency or length of trial) had a higher likelihood of adverse events. However, upon looking at the characteristics of the adverse events, it was found that only 15 adverse events were defined as related to telerehabilitation from all the studies that reported relatedness. In addition, for the adverse events with severity information available, 87.4% were defined as non-serious, mild, or minor.

A slightly larger proportion of asynchronous telerehabilitation studies had adverse events (9/30 studies, 30.0%), in comparison to synchronous telerehabilitation studies (13/46 studies, 28.3%). In addition, the total number of adverse events from asynchronous telerehabilitation studies represented a larger proportion of total adverse events from all included studies. This could be due to other characteristics of the asynchronous telerehabilitation studies, like disease type or the criteria used to define adverse events. However, it is also possible that due to the lack of real-time monitoring, asynchronous telerehabilitation studies allowed for more opportunity for adverse events to occur. More research is needed to compare the safety of the two methods of telerehabilitation, as asynchronous telerehabilitation provides certain benefits over synchronous telerehabilitation. For instance, it does not rely on real-time consultations which require the patient and provider to schedule a time both parties are available, and thus may improve the accessibility of rehabilitation services [112].

Categorized by disease, most adverse events were from cardiac studies, with 92 events across 6 studies, Parkinson's disease studies with 75 adverse events across 3 studies, and a multiple sclerosis study, with 60 adverse events from 1 study [23, 26, 28, 29, 31, 32, 82, 83, 86, 91]. It is unclear if the adverse events can be attributed to telerehabilitation specifically, or if the nature of these participants' diseases simply predisposes them to more adverse events whether at rest, during telerehabilitation, or during in-person rehabilitation. For example, of the two cardiac studies that defined what was classified as an adverse event, they included cardiac symptoms such as palpitations and angina, which can occur at rest in cardiac populations [29, 32, 113, 114]. To support the idea that certain diseases cause a predisposition to adverse events, one of these cardiac studies had a non-telerehabilitation control group and reported no significant difference in the number of adverse events between the two groups [32]. In addition, the criteria for what constitutes an adverse event varied from study to study, which could explain the variation in the number of adverse events reported. For instance, one of the Parkinson's disease studies included baseline pain, 'not feeling well', and medication side effects as adverse events, which are common day-to-day events regardless of participation in rehabilitation [82, 115]. Many studies did not specifically define what constituted an adverse event.

Adverse events were mostly physical, with only 3 adverse events being non-physical. However, once again, it was unclear what qualified as an adverse event. One study reported a fear of the possibility of injury or falls. However, the study did not provide any detail as to what specifically about the participant's fear met the criteria of an adverse event. Nevertheless, this finding points to the need to add safety measures and technical support to ease patients' fears with technology when implementing telerehabilitation. These findings are consistent with other studies identifying a common lack of familiarity, fear, and frustration with digital technology during telerehabilitation [116].

Many studies were missing descriptive information on the intervention or details about the adverse events. Out of the 81 included studies, the total number of telerehabilitation sessions

was not reported in 3 studies, session frequency per week was not reported by 11 studies, length of intervention was not reported by 6 studies, and session length was not reported by 28 studies. There were 7 studies with adverse events that did not report the severity of the adverse events. There was 1 study that just described the types of adverse events but did not report the total number. In addition, of those reporting the severity of adverse events, 2 studies only reported the number of participants that experienced adverse events of each severity category, rather than specific number of adverse events. Detailed reporting is important for the replicability of studies, as well as analysis of the safety of the telerehabilitation interventions [21, 117].

This review has a number of strengths. For instance, all study phases (screening, extraction, synthesis) were completed in duplicate and independently. This review was guided by the JBI methodological framework for scoping reviews, followed recommendations for conducting a high-quality scoping review, and adhered to PRISMA-ScR guidelines [17, 18]. Our team included members with extensive experience in telerehabilitation and conducting scoping reviews.

Limitations of the review include the exclusion of gray literature and studies not published in the English language. No risk of bias assessment, or estimation and comparison of measurement errors were conducted, consistent with the PRISMA-ScR guidelines, as the aim of a scoping review is to map out the extent of literature rather than assess the quality of studies [17, 18]. Indeed, we have previously identified the need for patient training on platforms such as Zoom or Skype, in order to conduct virtual care [108].

This scoping review provides useful insights at an important juncture in time. There has been a surge in the popularity of telerehabilitation due to the COVID-19 pandemic, and now patients, providers, funders, and governments are contemplating how rehabilitation should be delivered in the future. Understanding adverse events related to telerehabilitation will help to identify key limitations for optimizing telerehabilitation delivery, by allowing for the development of necessary risk-mitigation measures and quality indicators. In particular, the finding that most adverse events were from specific patient populations such as Parkinson's disease, cardiovascular conditions, or multiple sclerosis, may suggest that these groups would benefit from more risk mitigation strategies when participating in telerehabilitation. This could include having a caretaker supervise the telerehabilitation session, providing a telerehabilitation kit [118] explaining important safety measures, or having emergency contacts in case of unexpected severe adverse events. A greater understanding of the safety and optimization of telerehabilitation could help influence government funding and guide policymakers, which is important as a lack of leadership and organizational support has been found to hinder the implementation of telerehabilitation [108]. With the COVID-19 pandemic, many organizations were forced to transition to virtual care delivery, with changes in the rules, regulations, and reimbursement models [119]. Understanding that the current literature shows adverse events are rare during telerehabilitation delivery could reassure policymakers and leaders about using telerehabilitation on an ongoing basis and encourage them to provide funding to grow these programs beyond the COVID-19 pandemic. Similarly, from a patient perspective, these findings of rare adverse events may aid in decision-making regarding participation in telerehabilitation.

Future directions of this scoping review may include a systematic review comparing the effect of telerehabilitation versus in-person care on adverse events, as this scoping review identified 28 randomized controlled trials that met inclusion criteria. Only 7 of the 28 randomized controlled trials identified any adverse events, but a formal systematic review would have to be completed. More studies should examine the role of non-physical adverse events during the delivery of telerehabilitation. Further studies should be conducted in pediatric populations, as there were limited studies identified.

In conclusion, this scoping review found that across the included studies, telerehabilitation was delivered with related adverse events being rare (0.3%), and mostly mild/non-severe. A comparable occurrence of adverse events (~30%) was found between asynchronous and synchronous telerehabilitation studies. When categorized by disease type, cardiac rehabilitation studies had the most frequent number of adverse events. Detailed reporting of interventions and adverse event characteristics is recommended for future studies. Telerehabilitation has grown in popularity and has the potential to make rehabilitation services more accessible to patients; however, more evidence on the safety of telerehabilitation is needed.

## Supporting information

**S1 Appendix. Preferred reporting items for systematic reviews and meta-analyses extension for scoping reviews (PRISMA-ScR) checklist.**
(PDF)

**S2 Appendix. MEDLINE(R) ALL 1946 to June 22, 2023, search strategy.**
(PDF)

**S3 Appendix. Excluded studies.**
(DOCX)

**S4 Appendix. Data extractors and date of extraction.**
(DOCX)

**S5 Appendix. Extraction table of eligible studies.**
(PDF)

**S6 Appendix. Prisma-ScR flow diagram.**
(DOCX)

## Acknowledgments

We would like to thank Thomasin Adams-Webber, Healthsearch, Library and Information Services, University Health Network, Toronto, Ontario, Canada, for the development of the search strategies, and Emilia Main, Healthsearch, Library and Information Services, University Health Network, Toronto, Ontario, Canada for the PRESS review.

## Author Contributions

**Conceptualization:** McKyla McIntyre, Angie Andreoli, Carl Froilan D. Leochico, Mark Bayley, Ailene Kua, Meiqi Guo, Sarah Munce.

**Data curation:** Thomas Yau, Josh Chan, Damanveer Bhogal, Angie Andreoli, Carl Froilan D. Leochico, Meiqi Guo.

**Formal analysis:** Thomas Yau, Josh Chan, Damanveer Bhogal, Angie Andreoli, Carl Froilan D. Leochico, Meiqi Guo.

**Funding acquisition:** McKyla McIntyre.

**Methodology:** Thomas Yau, Josh Chan, McKyla McIntyre, Damanveer Bhogal, Angie Andreoli, Carl Froilan D. Leochico, Mark Bayley, Ailene Kua, Meiqi Guo, Sarah Munce.

**Project administration:** Ailene Kua, Meiqi Guo.

**Supervision:** Meiqi Guo, Sarah Munce.

**Writing – original draft:** Thomas Yau, Josh Chan.

**Writing – review & editing:** Thomas Yau, McKyla McIntyre, Angie Andreoli, Carl Froilan D. Leochico, Mark Bayley, Ailene Kua, Meiqi Guo, Sarah Munce.

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
