## [Decision Letter · Decision Letter 0]

16 Jul 2024

PONE-D-24-06105Adverse events associated with the delivery of telerehabilitation: A scoping reviewPLOS ONE

Dear Dr. Munce,

Thank you for submitting your manuscript to PLOS ONE. After careful consideration, we feel that it has merit but does not fully meet PLOS ONE’s publication criteria as it currently stands. Therefore, we invite you to submit a revised version of the manuscript that addresses the points raised during the review process.

In particular:

- objectives and related conclusions should be made more explicit,

- language and typographical errors should be removed,

- literature review should include more references to relevant studies,

- format of reporting numerical values (e.g. percentages) should be made uniform,

- all abbreviations (also used in figures) should be defined (e.g. MSK in Fig. 3.),

- the flow diagram of systematic review would increase the readability of the manuscript.

We look forward to receiving your revised manuscript.

Kind regards,

Maciej Huk, Ph.D.

Academic Editor

PLOS ONE

“Author TY received funding through a Toronto Rehabilitation Institute summer research position, funded by a donation to the University Health Network.”

Reviewers' comments:

Reviewer's Responses to Questions

**Comments to the Author**

1. Is the manuscript technically sound, and do the data support the conclusions?

Reviewer #1: Yes

Reviewer #2: Yes

2. Has the statistical analysis been performed appropriately and rigorously? 

Reviewer #1: Yes

Reviewer #2: N/A

3. Have the authors made all data underlying the findings in their manuscript fully available?

Reviewer #1: Yes

Reviewer #2: Yes

4. Is the manuscript presented in an intelligible fashion and written in standard English?

Reviewer #1: Yes

Reviewer #2: Yes

5. Review Comments to the Author

Reviewer #1: The authors have provided sound rationale for the need for this research. They closely followed published recommendations for the design and reporting of scoping reviews. I have highlighted a few things in the pdf that need clarification.

Reviewer #2: Line 148: EBSCOhost to be written as EBSCOhost

Line 162: the link/reference to Covidence is to be provided.

Line 279-348 & Line 341: references are to be cited for the number of studies mentioned.

Line 332-336: the total adverse events are to be stated.

S3. Appendix. Extraction table: Numerous typographical errors required revision. Please use word checking or Grammarly software to detect errors.

S3 Appendix. Extraction table: all the references are to cited under the first column 'title' and cross cited in the text where applicable.

S4 Appendix: the word wrong is to be replaced with a more suitable word.

Decimal points for percentage figures are to be standardized.

Reference 12: PM R is to be spelled out.

6. PLOS authors have the option to publish the peer review history of their article (what does this mean?). If published, this will include your full peer review and any attached files.

Reviewer #1: No

Reviewer #2: No

---

## [Author Response · Author response to Decision Letter 0]

31 Jul 2024

July 31, 2024

Subject: Revision submission of our revised manuscript, “Adverse events associated with the delivery of telerehabilitation across rehabilitation populations: A scoping review” for consideration for publication in PLOS ONE.

Dear Reviewers:

Please find enclosed the revised manuscript, “Adverse events associated with the delivery of telerehabilitation across rehabilitation populations: A scoping review” for consideration for publication as a scoping review in PLOS ONE. We would like to thank you for the opportunity to resubmit a revised copy of this manuscript. We would also like to take this opportunity to express our thanks to the reviewers for the positive feedback and helpful comments for correction or modification. The manuscript has been revised to address the reviewers’ comments which are marked using track changes. We also provided our response to their comments below. The revisions have been developed in consultation with all coauthors, and each author has given approval to the final form of this revision. 

Responses to Reviewers’ Comments

Reviewer 1

Comment 1: According to Peters et al (2020) it is recommended that key elements of the inclusion criteria are reflected in the title. They suggested the "PCC" mnemonic (population, concept, and context) as a guide to selecting the title. Please consider doing this. 

Response 1: Thank you for pointing this out. We have ensured our title and aim now reflect the PCC framework. P=Across rehabilitation populations, Concept=Adverse events [phenomenon of interest/outcomes], Context=Telerehabilitation [setting].

New title: “Adverse events associated with the delivery of telerehabilitation across rehabilitation populations: A scoping review.” (page 1)

New aim “This scoping review aimed to conduct a systematic search of published literature on adverse events during the delivery of telerehabilitation, across rehabilitation populations, and map out the extent of existing research. This included identifying characteristics of adverse events (frequency, physical versus non-physical, relatedness, severity) and examining adverse events for different modes of telerehabilitation delivery and disease states. (page 6)

Comment 2: I would like to see this objective more explicit

Response 2: Thank you for the comment. We have adjusted the objective statement to be more explicit as follows: “Objective: This scoping review aimed to map existing research on adverse events encountered during telerehabilitation delivery, across rehabilitation populations. This includes identifying characteristics of adverse events (frequency/physical/non-physical, relatedness, severity) and examining adverse events by different modes of telerehabilitation delivery and disease states.” (page 3)

Comment 3: If you make your study objective(s) more explicit, the make sure that the reported results (likely the most important results) and conclusion related back to the study's objective(s).

Response 3: Thank you for the suggestion. We have ensured that the results and conclusion reflect the study’s objectives. Please see below. 

Results: 

- Frequency (page 15): “A total of 295 adverse events occurred during 84,534 telerehabilitation sessions, representing 0.31%. Of the 81 included studies, 59 reported no adverse events.”

- Physical versus non-physical (page 17): “Almost all the adverse events were physical. The majority were pain-related (musculoskeletal pain/strain or headache), but also included fatigue, falls, dizziness, and cardiac-related adverse events (e.g., chest discomfort, palpitations, angina, tachypnea, etc.). Only 3 adverse events were non-physical. For instance, one participant felt concerned about the possibility of falling, another participant had an unrelated adverse event of anxiety, and a patient with depression felt demotivated when asked to complete a measure of mood [48, 64, 93].”

- Relatedness (page 16,17): “Only 15 of the 295 total adverse events were characterized by study authors as related, 2 probably related, 8 possibly related, 1 potentially related, 5 unlikely, 147 unrelated, and 94 undefined. There were 2 studies that only reported the number of participants per level of relatedness, rather than the specific number of adverse events. Another study reported a description of the adverse events but without a numerical count or report of relatedness.”

- Severity (page 16): “The severity of the adverse events was mostly classified as non-serious, mild, or minor by the study authors (Table 1).”

- Different modes of telerehabilitation delivery (page 13-16): some excerpts include: “There were 63 studies that were fully telerehabilitation-based (i.e., patients and therapists in separate physical locations) [25-27, 29-32, 34-37, 40-43, 45-50, 52-55, 57-59, 61, 63, 64, 66, 68-71, 73, 76-85, 87-98, 100-102, 104], and 18 were hybrid (i.e., with an in-person component) [24, 28, 33, 38, 39, 44, 51, 56, 60, 62, 65, 67, 72, 74, 75, 86, 99, 103]. There were 46 studies that used synchronous (real-time) methods of delivering telerehabilitation [24, 29-32, 34, 35, 38-44, 46-52, 55-57, 62-65, 69, 71, 74-76, 81-85, 87, 89, 92, 94, 95, 97, 101, 102], 30 studies that used asynchronous (store-and-forward) methods [25-28, 33, 36, 37, 53, 54, 58-61, 66, 70, 72, 73, 77, 78, 86, 88, 90, 91, 93, 96, 98-100, 103, 104], and 5 studies that were hybrid [45, 67, 68, 79, 80]. On average, sessions were provided 2.68 times per week, for 10.26 weeks, and for 46.52 minutes per session. Various communication methods were used: videoconferencing (52 studies) [24, 27, 29-32, 34, 35, 38-41, 43-52, 55-57, 62-65, 67-69, 71, 73, 75, 76, 79-85, 87, 89, 92, 94, 95, 97, 101-103], phone calls (25) [25, 26, 28, 33, 36, 37, 53, 59, 60, 70, 72, 73, 75-79, 81, 86, 88, 93, 96, 98, 100, 104], mobile application (10) [34, 42, 53, 54, 59, 66, 74, 76, 99, 103], email (6) [54, 58, 72, 73, 77, 91], Internet-based virtual reality (VR) systems (3) [61, 67, 93], and text messaging (4) [36, 58, 79, 98].” 

“Of the 81 included studies, 40.0% of total adverse events (n=118) were obtained from those that employed synchronous telerehabilitation. Among those studies using synchronous telerehabilitation, 28.2% indicated some adverse events. In contrast, 60.0% of total adverse events (n=177) were obtained from studies that employed asynchronous telerehabilitation. Of those studies using asynchronous telerehabilitation, 30.0% indicated some adverse events. No adverse events were reported from those that employed hybrid telerehabilitation.”

- Disease states (page 17): “When categorized by disease type, cardiac rehabilitation studies had the most frequent adverse events, with 92 adverse events across 6 studies, including fatigue, palpitations, angina, diaphoresis, dyspnea, and syncope [24, 26, 28, 29, 31, 32]. Studies on people with Parkinson’s disease had the second most, with 75 adverse events across 3 studies, such as pain, neuropathy, loss of balance, dizziness, falls, and increased awareness of hand tremors [82, 83, 86]. Studies on multiple sclerosis were third, with 60 adverse events from 1 study, including falls and skin reactions [91].”

Conclusion:

- A line on synchronous versus asynchronous telerehabilitation has been added to the conclusion for “different modes of telerehabilitation delivery.” A line on disease type has been added for “disease states.” 

- (Page 24) “In conclusion, this scoping review found that across the included studies, telerehabilitation was delivered with related adverse events being rare, and mostly mild/non-severe. A similar percentage of asynchronous studies reported adverse events with telerehabilitation delivery in comparison to synchronous studies. When categorized by disease type, cardiac rehabilitation studies had the most frequent adverse events. Detailed reporting of interventions and adverse event characteristics is recommended for future studies. Telerehabilitation has grown in popularity and has the potential to make rehabilitation services more accessible to patients; however, more evidence on the safety of telerehabilitation is needed.”

- (Abstract Page 4): “Conclusions: From the 81 included studies, telerehabilitation was delivered with related adverse events being rare, and mostly characterized as mild/non-severe. A comparable occurrence of adverse events (~30%) was found between asynchronous and synchronous telerehabilitation studies. When categorized by disease type, cardiac telerehabilitation studies had the most frequent adverse events. Detailed reporting of telerehabilitation interventions and adverse event characteristics is recommended for future studies (i.e., use of TIDieR reporting guidelines). Telerehabilitation has the potential to make rehabilitation services more accessible to patients; however, more evidence on the safety of telerehabilitation is needed. 

Comment 4: should be "aimed" since the work has been completed and you are making a report on what was done and what you learned. Please re-edit for all word tense errors and inconsistencies.

Response 4: Thank you for your comment. We have made this suggested change throughout the manuscript.

Comment 5: Please make more explicit the questions and objectives you intend to address with reference to their elements, e.g., population or participants, concepts, and context or other relevant key elements.

Note: the page numbers in S1 do to appear to correspond consistently with the items on the checklist.

Response 5: Thank you for the comment. We hope we made the objective and aim statement more explicit as per above. S1 has been revised such that page numbers correspond with items on checklist. 

Objective (page 3):

“Objective: This scoping review aimed to map existing research on adverse events encountered during telerehabilitation delivery, across rehabilitation populations. This includes identifying characteristics of adverse events (frequency/physical/non-physical, relatedness, severity) and examining adverse events by different modes of telerehabilitation delivery and disease states.”

Aim (page 6): 

“This scoping review aimed to conduct a systematic search of published literature on adverse events during the delivery of telerehabilitation, across rehabilitation populations, and map out the extent of existing research. This included identifying characteristics of adverse events (frequency, physical versus non-physical, relatedness, severity) and examining adverse events for different modes of telerehabilitation delivery and disease states.”

Comment 6: in Appendix 3 your abbreviate quasi-experimental as quasi (please indicate the use of that abbrev here, e.g., quasi-experimental (quasi)

Response 6: Thank you, this has been revised. 

Comment 7: you used one abbrev in this figure, i.e., MSK. Please define this abbrev in the figure caption

Response 7: Thank you, the abbreviation has been defined in the figure caption. 

Comment 8: virtual reality (VR); since some readers may not know this abbrev

Response 8: Thank you for the comment. We have revised.

Comment 9: if possible, please provide a brief listing of specific types of adverse events associated with cardiac rehab, Parkinson's and MS patients

Response 9: Thank you for the suggestion. The paragraph has been revised to include examples of specific types of adverse events associated with these conditions.

 “When categorized by disease type, cardiac rehabilitation studies had the most frequent adverse events, with 92 adverse events across 6 studies, including fatigue, palpitations, angina, diaphoresis, dyspnea, and syncope [22, 24, 26, 27, 29, 30]. Studies on people with Parkinson’s disease had the second most, with 75 adverse events across 3 studies, such as pain, neuropathy, loss of balance, dizziness, falls, and increased awareness of hand tremors [80, 81, 84]. Studies on multiple sclerosis were third, with 60 adverse events from 1 study, including falls and skin reactions [89].” (page 17)

Comment 10: You checked off item 18 from the checklist but since your objectives in the introduction were not explicit it is not possible to know how your synthesis of results relates to study objectives.

Response 10: Thank you for the comment. We hope that with a more explicit objective statement, it is more clear as to how the synthesis of results relates to the objectives. Please see response 3 and 5.

Reviewer 2

Comment 1: Line 148: EBSCOhost to be written as EBSCOhost

Response 1: Thank you for your comment. We have revised to “EBSCOHost platform.”

Comment 2: Line 162: the link/reference to Covidence is to be provided.

Response 2: Thank you, this has been revised to the following: “They were then transferred into Covidence (https://www.covidence.org/), a web-based reference manager software.”

Comment 3: Line 332-336: the total adverse events are to be stated.

Response 3: Thank you, these lines have been revised to include total adverse events: 

“Only 15 of the 295 total adverse events were characterized by study authors as related, 2 probably related, 8 possibly related, 1 potentially related, 5 unlikely, 147 unrelated, and 94 undefined.” (page 16)

Comment 4: S3. Appendix. Extraction table: Numerous typographical errors required revision. Please use word checking or Grammarly software to detect errors.

Response 4: Thank you for pointing this out, Grammarly software was used to detect and revise the typographical errors throughout the manuscript. 

Comment 5: S3 Appendix. Extraction table: all the references are to cited under the first column 'title' and cross cited in the text where applicable.

Response 5: Thank you for the suggestion. References have been cited under the first column. They have also been cited in the text as appropriate. 

Comment 6: S4 Appendix: the word wrong is to be replaced with a more suitable word.

Response 6:

Thank you for the suggestion. The exclusion reasons have been more clearly described with the word “wrong” removed as seen below: 

Outcomes were not relevant (n = 24)

Intervention was not telerehabilitation (n = 33)

Ineligible study design (n = 35)

Not written in English language (n = 1)

No therapist involved (n = 4)

Comment 7: Decimal points for percentage figures are to be standardized.

Response 7: Thank you for pointing this out, all percentage figures have now been standardized to one decimal point. 

Comment 8: Reference 12: PM R is to be spelled out.

Response 8: Thank you for the suggestion. It has now been spelled out to be Physical Medicine & Rehabilitation.

We hope that we have satisfactorily addressed the reviewers’ comments. Please do not hesitate to contact us if you require any further information. Thank you and have a wonderful day.

Kind Regards,

Sarah Munce, PhD on behalf of the Team

Bloorview Research Institute, Holland Bloorview Kids Rehabilitation Hospital & Institute of Health Policy, Management and Evaluation, Rehabilitation Sciences Institute

E-mail: sarah.munce@uhn.ca

---

## [Decision Letter · Decision Letter 1]

2 Sep 2024

PONE-D-24-06105R1Adverse events associated with the delivery of telerehabilitation across rehabilitation populations: A scoping reviewPLOS ONE

Dear Dr. Munce,

Thank you for submitting your manuscript to PLOS ONE. After careful consideration, we feel that it has merit but does not fully meet PLOS ONE’s publication criteria as it currently stands. Therefore, we invite you to submit a revised version of the manuscript that addresses the points raised during the review process.

In particular:language problems should be corrected,the identified problem and rationale for conducted study should be presented explicitly,not needed column in Appendix S3 table should be removed,estimation and comparison of measurement errors should be presented or lack of it should be listed as limitation of the study.

We look forward to receiving your revised manuscript.

Kind regards,

Maciej Huk, Ph.D.

Academic Editor

PLOS ONE

Journal Requirements:

Reviewers' comments:

Reviewer's Responses to Questions

**Comments to the Author**

1. If the authors have adequately addressed your comments raised in a previous round of review and you feel that this manuscript is now acceptable for publication, you may indicate that here to bypass the “Comments to the Author” section, enter your conflict of interest statement in the “Confidential to Editor” section, and submit your "Accept" recommendation.

Reviewer #1: All comments have been addressed

Reviewer #2: (No Response)

2. Is the manuscript technically sound, and do the data support the conclusions?

Reviewer #1: Yes

Reviewer #2: Yes

3. Has the statistical analysis been performed appropriately and rigorously? 

Reviewer #1: Yes

Reviewer #2: N/A

4. Have the authors made all data underlying the findings in their manuscript fully available?

Reviewer #1: Yes

Reviewer #2: Yes

5. Is the manuscript presented in an intelligible fashion and written in standard English?

Reviewer #1: Yes

Reviewer #2: Yes

6. Review Comments to the Author

Reviewer #1: Thank you for addressing previous comments. I have only a few very minor comments, which were made in the corrected version of the manuscript (see pdf), and listed here:

lines 103-104 - this statement is true assuming that appropriate internet connections are available; you might make this qualification

line 214 - the word should be "was" not "is"

line 392 - Also, please briefly restate the problem you identified and the rational/need for this study.

line 451 - instead of stating "most" please provide an explicit percentage.

line 553 - suggest: "...most frequent number of..."

Reviewer #2: For S3 Appendix, it is sufficient to list only the authors (e.g., 'Lundgren et al.') as in the second column. Just need to include the reference number that was cited in the text after the authors' name e.g. Lundgren et al. [ ] in the second column. First column is to be omitted.

7. PLOS authors have the option to publish the peer review history of their article (what does this mean?). If published, this will include your full peer review and any attached files.

Reviewer #1: No

Reviewer #2: No

---

## [Author Response · Author response to Decision Letter 1]

12 Sep 2024

September 12, 2024

Subject: Revision submission of our revised manuscript, “Adverse events associated with the delivery of telerehabilitation across rehabilitation populations: A scoping review” for consideration for publication in PLOS ONE.

Dear Reviewers:

Please find enclosed the revised manuscript, “Adverse events associated with the delivery of telerehabilitation across rehabilitation populations: A scoping review” for consideration for publication as a scoping review in PLOS ONE. We would like to thank you for the opportunity to resubmit a revised copy of this manuscript. We would also like to take this opportunity to express our thanks to the reviewers for the positive feedback and helpful comments for correction or modification. The manuscript has been revised to address the reviewers’ comments which are marked using track changes. We also provided our response to their comments below. The revisions have been developed in consultation with all coauthors, and each author has given approval to the final form of this revision. 

Responses to Reviewers’ Comments

Reviewer 1

Comment 1: Telerehabilitation can also provide services to those who would not normally be

able to access traditional rehabilitation, such as those living in remote communities or patients

with disabilities which hinder participation in in-person sessions. This is assuming that appropriate internet connections are available

Response 1: Thank you for the comment. The revised sentence now reads: “Telerehabilitation can also provide services to those who would not normally be able to access traditional rehabilitation, such as those living in remote communities or patients with disabilities which hinder participation in in-person sessions [2], assuming that appropriate internet connections are available.” (page 5)

Comment 2: Quality/risk of bias assessment was not completed as this is not the purpose of a scoping review. It should be “was”.

Response 2: Thank you for the comment. The revised sentence now reads: “Quality/risk of bias assessment was not completed as this was not the purpose of a scoping review.” (page 9)

Comment 3: The purpose of this scoping review was to map the existing research on adverse events during the delivery of telerehabilitation. Also, please briefly restate the problem you identified and the rational/need for this study.

Response 3: 

Thank you for the suggestion. “The purpose of this scoping review was to map the existing research on adverse events during the delivery of telerehabilitation” has been rephrased to include the full aim statement and problem identified. “There has yet to be any synthesis of the existing literature that summarizes the currently available research on adverse events related to telerehabilitation. This scoping review aimed to conduct a systematic search of published literature on adverse events during the delivery of telerehabilitation, across rehabilitation populations, and map out the extent of existing research. This included identifying characteristics of adverse events (frequency, physical versus non-physical, relatedness to telerehabilitation, severity) and examining adverse events for different modes of telerehabilitation delivery and disease states.” (page 17)

Comment 4: In addition, most of the adverse events were defined as non-serious, mild, or minor. instead of stating "most" please provide an explicit percentage.

Response 4: Thank you for the comment. The sentence has been revised to be more explicit with a percentage. “In addition, for the adverse events with severity information available, 87.4% were defined as non-serious, mild, or minor.” (page 20)

Comment 5: When categorized by disease type, cardiac rehabilitation studies had the most frequent adverse events. Add “number of” adverse events.

Response 5: Thank you for the comment. The revised sentence now reads: “When categorized by disease type, cardiac rehabilitation studies had the most frequent number of adverse events.” (page 24)

Reviewer 2

Comment 1: Language problems should be corrected

Response 1: Thank you for the comment. The paper has been run through the Grammarly software and any identified language problems have been corrected. 

Comment 2: The identified problem and rationale for conducted study should be presented explicitly

Response 2: Thank you for the comment. As per above, the discussion now starts with the full aim statement and problem identified. This is also reflected in the abstract and introduction sections. The rationale for the current review is as follows: “There has yet to be any synthesis of the existing literature that summarizes the currently available research on adverse events related to telerehabilitation. Thus, this scoping review aimed to conduct a systematic search of published literature on adverse events during the delivery of telerehabilitation, across rehabilitation populations, and map out the extent of existing research. This included identifying characteristics of adverse events (frequency, physical versus non-physical, relatedness to telerehabilitation, severity) and examining adverse events for different modes of telerehabilitation delivery and disease states.” (page 17)

Comment 3: not needed column in Appendix S3 table should be removed. 

Response 3: Thank you for the comment. Appendix S3 table’s extra column has been removed, and remaining column edited to include reference number as per the following instructions: “For S3 Appendix, it is sufficient to list only the authors (e.g., 'Lundgren et al.') as in the second column. Just need to include the reference number that was cited in the text after the authors' name e.g., Lundgren et al. [ ] in the second column. First column is to be omitted.”

Comment 4: estimation and comparison of measurement errors should be presented or lack of it should be listed as limitation of the study

Response 4: Thank you for the suggestion. The limitations section has been revised to include this: No risk of bias assessment, or estimation and comparison of measurement errors were conducted, consistent with the PRISMA-ScR guidelines, as the aim of a scoping review is to map out the extent of literature rather than assess the quality of studies [17, 18].

We hope that we have satisfactorily addressed the reviewers’ comments. Please do not hesitate to contact us if you require any further information. Thank you and have a wonderful day.

Kind Regards,

Sarah Munce, PhD on behalf of the Team

Bloorview Research Institute, Holland Bloorview Kids Rehabilitation Hospital & Institute of Health Policy, Management and Evaluation, Rehabilitation Sciences Institute

E-mail: sarah.munce@uhn.ca

---

## [Decision Letter · Decision Letter 2]

24 Oct 2024

Adverse events associated with the delivery of telerehabilitation across rehabilitation populations: A scoping review

PONE-D-24-06105R2

Dear Dr. Munce,

We’re pleased to inform you that your manuscript has been judged scientifically suitable for publication and will be formally accepted for publication once it meets all outstanding technical requirements.

Kind regards,

Maciej Huk, Ph.D.

Academic Editor

PLOS ONE

Additional Editor Comments (optional):

Reviewers' comments:

Reviewer's Responses to Questions

**Comments to the Author**

1. If the authors have adequately addressed your comments raised in a previous round of review and you feel that this manuscript is now acceptable for publication, you may indicate that here to bypass the “Comments to the Author” section, enter your conflict of interest statement in the “Confidential to Editor” section, and submit your "Accept" recommendation.

Reviewer #1: All comments have been addressed

Reviewer #2: All comments have been addressed

2. Is the manuscript technically sound, and do the data support the conclusions?

Reviewer #1: Yes

Reviewer #2: (No Response)

3. Has the statistical analysis been performed appropriately and rigorously? 

Reviewer #1: Yes

Reviewer #2: (No Response)

4. Have the authors made all data underlying the findings in their manuscript fully available?

Reviewer #1: Yes

Reviewer #2: (No Response)

5. Is the manuscript presented in an intelligible fashion and written in standard English?

Reviewer #1: Yes

Reviewer #2: (No Response)

6. Review Comments to the Author

Reviewer #1: Thank you for addressing earlier comments. This manuscript has improved. I have no other questions or concerns.

Reviewer #2: (No Response)

7. PLOS authors have the option to publish the peer review history of their article (what does this mean?). If published, this will include your full peer review and any attached files.

Reviewer #1: No

Reviewer #2: No

---

## [Editor Report · Acceptance letter]

8 Nov 2024

PONE-D-24-06105R2 

PLOS ONE

Dear Dr. Munce, 

I'm pleased to inform you that your manuscript has been deemed suitable for publication in PLOS ONE. Congratulations! Your manuscript is now being handed over to our production team.

Kind regards, 

on behalf of

Dr. Maciej Huk 

Academic Editor

PLOS ONE